# Using refined methods to pick up mice: A survey benchmarking prevalence & beliefs about tunnel and cup handling

**Lauren Young[1], Donna Goldsteen[2], Elizabeth A. Nunamaker[3], Mark J. Prescott[4], Penny Reynolds[5], Sally Thompson-Iritani[6], Sarah E. Thurston[3], Tara L. Martin[7], Megan R. LaFollette[8]***

**1** Department of Integrative Biology, University of Guelph, Guelph, Ontario, Canada, **2** Independent Consultant (Formerly AstraZeneca), Damascus, Maryland, United States of America, **3** Charles River Laboratories, Global Animal Welfare and Training, Wilmington, Massachusetts, United States of America, **4** National Centre for the Replacement, Refinement and Reduction of Animals in Research (NC3Rs), London, United Kingdom, **5** University of Florida, Gainesville, Florida, United States of America, **6** University of Washington, Seattle, WA, United States of America, **7** Refinement and Enrichment Advancements Laboratory, University of Michigan, Ann Arbor, Michigan, United States of America, **8** The 3Rs Collaborative (3RsC), Denver, Colorado, United States of America

* meglafollette@na3rsc.org

**Data Availability Statement:** Data are available via the following citation & URL: Martin, T. L., Young, L., Goldsteen, D., Nunamaker, E., Reynolds, P.,

## Abstract

Refined handling improves laboratory mouse welfare and research outcomes when compared to traditional tail handling, yet implementation does not seem to be widespread. Refined handling includes picking up a mouse using a tunnel or cupped hands. The aim of this study was to determine the current prevalence of and beliefs towards refined handling using the theory of planned behavior. It was predicted that refined handling prevalence is low compared to traditional handling methods, and its implementation is determined by individual and institutional beliefs. Research personnel were recruited via online convenience sampling through email listservs and social media. A total of 261 participants in diverse roles (e.g. veterinarians, managers, caretakers, researchers, etc.) responded primarily from the USA (79%) and academic institutions (61%) Participants were surveyed about their current use, knowledge, and beliefs about refined handling. Quantitative data were analyzed via descriptive statistics and generalised regression. Qualitative data were analyzed by theme. Research personnel reported low levels of refined handling implementation, with only 10% of participants using it exclusively and a median estimate of only 10% of institutional mice being handled with refined methods. Individually, participants had positive attitudes, neutral norms, and positive control beliefs about refined handling. Participants' intention to provide refined handling in the future was strongly associated with their attitudes, norms, and control beliefs (p<0.01). Participants believed barriers included jumpy mice, perceived incompatibility with restraint, lack of time, and other personnel. However, participants also believed refined handling was advantageous to mouse welfare, handling ease, personnel, and research. Although results from this survey indicate that current refined handling prevalence is low in this sample, personnel believe it has important benefits, and future use is associated with their beliefs about the practice. People who believed refined handling was good, felt pressure to use it, and were confident in their use reported higher implementation.

Thompson-Iritani, S., Thurston, S., LaFollette, M. (2023). Dataset Refined Methods of Mouse Handling Survey [Data set], University of Michigan - Deep Blue Data. https://doi.org/10.7302/8bhs-2v35.

**Funding:** The author(s) received no specific funding for this work.

**Competing interests:** The authors have declared that no competing interests exist.

Increased refined handling could be encouraged through education on misconceptions, highlighting advantages, and addressing important barriers.

## Introduction

Laboratory animal welfare is impacted by a variety of factors that are inherent to the laboratory environment including housing, potential restriction of natural behaviors, and interactions with humans [1]. These factors can result in negative impacts on animals, ranging from a singular aversive experience to a lifetime of cumulative stressors [2]. These impacts can also be positive, especially as institutions strive to improve practices to promote animal welfare [3]. One important practice to focus on improving is animal handling. Research animals must be handled regularly for a variety of key tasks such as regular husbandry, health exams, and experimental procedures [4, 5]. These routine interactions have the potential to be experienced as either positive or negative by both human and animal depending on the quality of the interaction. Refined practices allow animal handlers the opportunity to prevent negative experiences and instead promote positive ones.

Traditionally, laboratory mice (*Mus musculus*) are moved by picking them up by the base of their tail, commonly referred to as "tail handling" [6]. However, handling of mice can be refined by instead picking them up using a tunnel or cupped hands, which we now term "refined handling". In this paper, "refined handling" refers only to how mice are picked up and can be followed by any type of restraint or typical procedure. Refined handling is beneficial to both laboratory animal welfare and research outcomes [7–18]. In fact, refined handling is often called low-stress or non-aversive handling because it decreases the stress and aversion the animal experiences when it is handled [7]. Other benefits of refined handling include reducing negative affective states in mice, such as anxiety and depression-like behavior [8–12], which in turn improves research outcomes [8, 13, 14, 15]. These benefits endure even after common procedures are performed (e.g., injections, oral gavage, anesthesia) which allows animal care staff and researchers to complete necessary tasks without extinguishing the positive effects of refined handling [6, 7, 8, 16]. Additionally, this refinement reduces the potentially confounding impacts of handling stress to improve the reliability and reproducibility of research data [17, 18].

Despite the important, well-known benefits of refined mouse handling and significant adoption in the United Kingdom where the methods originated [8] globally there seems to be an overall lack of adoption or implementation [19]. It is suspected that this is caused by many common factors. Previous studies on refinements to handling of mice and rats generally have indicated that participants perceive barriers to include time, personnel attitudes, lack of training, and research factors [19, 20]. Additionally in a 2019 survey, numerous misconceptions existed about the incompatibility of refined mouse handling with common procedures [19]. These beliefs may be preventing laboratory mice from receiving a refinement that would significantly improve their welfare, making it pertinent to investigate and target these perceived barriers. This survey is needed to provide updated benchmarking of prevalence, determine quantitative associations between implementation and potential factors, and to reassess potential advantages and barriers to current use for refined mouse handling specifically.

The theory of planned behavior is a tool used to explain human behavior and predict behavioral change [21]. This theory states that the likelihood of performing a particular behavior is most directly related to an individual's intention or plans to execute that behavior. In turn, the

intention to perform a behavior is impacted by three main beliefs: attitudes, control beliefs, and subjective norms. _Attitudes_ about executing a behavior refer to how an individual feels about the behavior, whether it is good, useful, right, and beneficial [22]. _Control_ beliefs refer to an individual's perceived confidence and ability to execute a behavior. Lastly, _subjective norms_ refer to societal or professional pressure on an individual to execute a behavior. By measuring these beliefs and their association to intentions, agencies can determine potential targets for interventions to increase the performance of behavior [22]. This method has been used in hundreds of published behavioral studies, including recent experimental surveys on the enactment of animal welfare improvements by animal care personnel [23].

Our aim in this study was to investigate levels of refined handling use and identify factors that prevent and/or enable its implementation. Our specific objectives were to gain insight into adoption of refined mouse handling, personnel knowledge and attitudes towards the method, and in particular, barriers to its implementation. Additionally, we evaluated each aim at both the institutional and individual levels, as refined handling often begins as an individual practice before becoming an institution-wide practice. Based on previous research, we hypothesized that current refined handling prevalence is low, and this is moderated by both individual and institutional factors.

## Materials and methods

All procedures, informational sheet, and online waived signed consent protocols were reviewed and determined to be exempt from IRB oversight and ongoing review by University of Michigan's Human Research Protection Program Institutional Review Board (IRB), protocol #HUM00195571. No interactions occurred between the researchers and mice during the course of the study, therefore approval from an Institutional Animal Care and Use Committee (IACUC) was unnecessary.

### Participants and procedures

Participants were recruited via online convenience sampling between April 27th and May 17th of 2021. Recruitment occurred using one of four contact methods: direct emails to known relevant research animal personnel, list serves (e.g., LAREF), email lists (e.g., NA3RsC) and social media communication (LinkedIn, Facebook, Twitter). Combined, this recruitment may have reached roughly up to 10,000 eligible participants. All modalities were contacted up to three times, with each subsequent contact containing slightly different text which is in line with recommended survey procedures [24]. Participants were included in the study if they were over the age of 18 and were laboratory staff working with mice; no other exclusion criteria were used. Participants were provided with an online information sheet to read prior to the study that assured them responses would be kept confidential and they were asked to document waived signed consent via a yes or no question. Then, participants completed a 10-minute online questionnaire administered via Qualtrics. Subsequent responses were kept anonymous and confidential; potentially identifiable information was only accessed by 3 core research team members.

**Measures.** This cross-sectional survey was created by members of the 3Rs Collaborative's Refinement Initiative using a review of the literature, the theory of planned behavior [25], and consultation with experts in the field of survey methodology and laboratory animal medicine. When possible, we used relevant validated survey instruments (e.g., theory of planned behavior survey [19]), but when these did not exist additional items were created, reviewed by our expert team, piloted by participants similar to the desired sample, and revised as necessary. Overall, participants answer 55 to 86 questions (depending on if they worked with mice

hands-on, hands-off, or both) across 4 major sections as described below All survey question text and scoring scales can be found in **S1 Table**.

To provide participants an opportunity to share their opinions freely, while simultaneously gaining insight into specific topics of our interest, we used a mixed methods approach of both qualitative and quantitative questions (i.e., open and close ended questions). When the survey was designed, we used the term 'non-aversive' handling, as this was the most accurate terminology for tunnel/cup handling at the time. We have since changed our terminology to 'refined handling', as we received feedback from our stakeholders that personnel were less receptive to the term 'non-aversive' in our outreach. For this reason, the term 'non-aversive' is used only in the methods section of this paper, to align with the exact questions asked in the survey.

**Demographics & work dactors.** Demographic information was obtained for age, current location, and highest education level. Participant gender was not obtained. Information on current work practice included location, type of institution (academic, industry, government, other), name of institution (if the participant felt comfortable sharing), role (e.g., veterinarian, manager, caretaker, researcher), number of years working with mice, whether mice were directly handled as part of their work, and if they were familiar with their institutions' policies on handling mice. We also asked participants for their email address to allow for a longitudinal survey collection which will be reported independently of these data. All potentially identifiable information (e.g., email address, institution name) was removed from data files prior to analysis to protect confidentiality.

**Measuring refined handling on individual & institutional levels.** Refined handling is a practice that must be applied by individual animal handlers and is often initially applied by a small group of individuals. However, changes at the institutional level in terms of policy and practice will create more widespread change. Additionally, some individuals may never (or rarely) directly handle laboratory mice but are essential to crafting institutional policies. To capture these two distinctly important levels of implementation, participants were asked to make assessments about refined mouse handling prevalence and beliefs on both an individual and institutional level depending on their knowledge and hands-on work with mice.

**Benchmarking refined handling.** To benchmark the frequency of refined handling, participants were asked to evaluate their current and planned mouse handling methods. First, participants were asked how familiar they are with non-aversive handling. 'Non-aversive handling' was defined to participants as "*the practice of picking up mice with a tunnel or with cupped hands (instead of picking them up via the tail or scruff)*." Of note, there are additional refinements to tail handling that were not covered in this survey, including picking up mice on a cage ladder [27] and cupping on the hands with massage [13]. Participants were also asked what methods they used to pick up mice (tail handling, forceps, tunnel handling, cupping, or other). Participants were asked what percentage of their current practice and future anticipated handling use non-aversive methods, how many cages of mice they work with per day, and a rough estimate of the number of mouse cages at the institution.

Next, participants were asked an open-ended question to determine what, if any, are the approved instances/exceptions when non-aversive handling could not be used to pick up mice at their institution. This question allowed us to gain insight into any major instances where refined handling could be considered impossible or extremely difficult. This would allow future initiatives to target more closely, and work to remove, these specific obstacles.

To benchmark accurate knowledge and potential misconceptions about refined handling, participants were asked a series of questions with correct or incorrect answers about the practice. These questions were developed by laboratory animal veterinarians, experts in refined handling, and based upon a thorough review of the literature. Questions specifically asked how

tail and refined handling related to mouse welfare, behavior, scientific outcomes, and its integration with routine procedures (restraint, injections, anesthesia).

At the end of the survey, we asked participants currently using refined handling what the single most important factor was that allowed them to begin using this method. These responses show us which current strategies for implementation are effective, so we can promote these changes in institutions that are currently working to implement refined handling.

**Beliefs: Theory of planned behavior.** The theory of planned behavior was used to assess refined handling intentions and beliefs, including behavioral attitudes, subjective norms and perceived behavioral control as described in the introduction of this paper. Surveys constructed using this theory typically have excellent reliability and validity [25].

First, participants were asked two qualitative, open-ended questions to allow them to answer freely without direct prompting from the study researchers. Participants were asked what they believe makes it difficult to use non-aversive handling for mice (e.g., barriers) and what advantages there are to using non-aversive handling.

Then, participants were asked a series of quantitative close-ended questions on a scale from 1–7 to assess their attitudes, norms, control beliefs, and intentions. Exact wording of these questions is included in **S1 Table**. These questions were developed from a manual on constructing theory of planned behavior questionnaires. Each construct was assessed by at least three items and summary variables were calculated based on their average. To further assess current beliefs, participants were also asked if they believe non-aversive handling is beneficial for animal welfare, for science and for people.

## Data analysis

**Quantitative analysis.** Data were analyzed using descriptive statistics and generalized linear regression. Descriptive statistics are presented as mean and standard deviation (SD), or median and interquartile range (IQR) for continuous data, and counts $n$ and percent (%) for count data. For summary scales, an average of individual items was collected for participants who answered at least 51% of subscale items. Duplicate survey responses were identified by matching email addresses and then the most complete response was retained.

For benchmarking institutional adoption of refined mouse handling, one participant per institution was designated as the primary respondent based on the following metrics. First, we included only individuals that listed an institution that either has only one site or who were associated with one particular site of a global institution. If there were multiple participants from an institution, then a primary institutional response was determined by the researchers using the following parameters. First the primary response was only chosen to be respondents that were familiar with institutional practices, with complete responses, and familiar with refined handling. Once met, the primary institutional response was determined based on role preferring training coordinators, over veterinarians, over managers, over researchers.

Generalized linear regression was used to determine the association between refined handling implementation and explanatory variables. Generalised regression models were fitted with normal canonical link function, and estimation and variable selection performed by adaptive lasso. To enhance prediction accuracy and interpretability of the models, variables were selected using the Least Absolute Shrinkage Selection Operator (LASSO). This is an extension of ordinary least squares regression where regression coefficients are estimated by adding a penalty function (the LASSO) to the residual sum of squares. The penalty function results in those regression coefficients that are least associated with the response variable being decreased to zero, which effectively removes them from the model. Variables were retained in the model if the P-value for the Wald chi-squared < 0.05. The dependent variables were

intention to implement refined handling at both the individual and institutional levels. The independent variables included theory of planned behavior beliefs, knowledge about refined mouse handling, and demographics. Residual plots were used to confirm residual distributional assumptions and model fit, and model selection by Akaike and Bayesian Information Criteria (AIC, BIC), and the generalized $R^2$. Data analysis was conducted in JMP Pro v 15. (SAS Institute Cary NC).

In the regression, demographic categories were collapsed into larger categories prior to analysis to assist with unbalanced and small sample sizes. Location was dichotomized into USA and Other. Institution was collapsed into the categories were Academia, Industry, and Other. Location was collapsed into categories (Veterinarian, Researcher, Manager, Technician, Caretaker, Other) that were entered into the model. The choice of which comparisons to drop from the final model was performed by the LASSO method described above with Veterinarian as the reference category.

This representative analysis was used: Intention to Implement Refined
Handling = Attitudes + Norms + Control Beliefs + Knowledge + Familiarity + Role + Country

**Qualitative analysis.** We used an inductive, bottom-up, content analysis to derive themes from open-ended participant responses. The complete coding manual with themes can be found in **S2 Table**. All coding and analyses were conducted in Microsoft Excel. The same manual was created and used to code both the individual and institutional responses for the same question category.

The goal of the coding process was to extract general themes from participant responses for subsequent analysis. The process is displayed in **Fig 1**. To summarize, one researcher (LEY) first read through all the written responses for a particular question category (e.g., Individual

A qualitative coding manual was created via a defined, iterative process

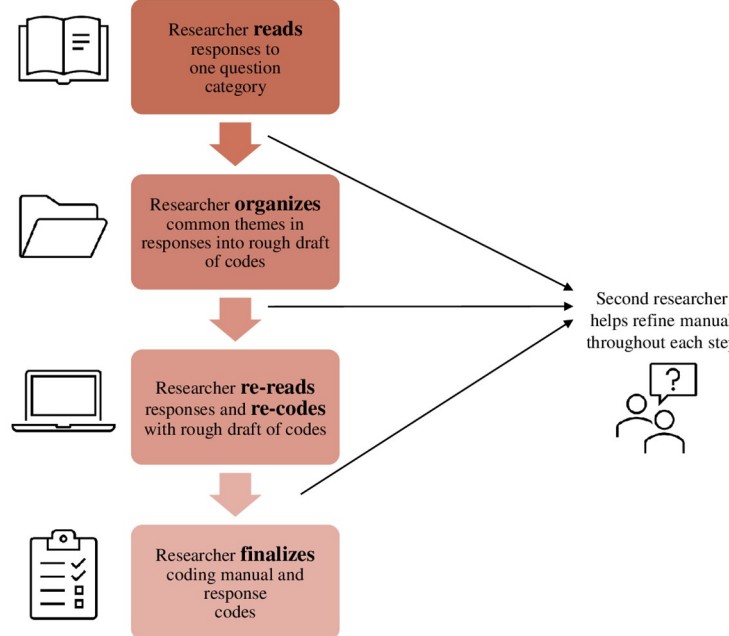

**Fig 1. Developing the coding manual with an iterative process.** Graphic outlines the formation of the coding manual and the process of coding the responses to each qualitative survey question category. One researcher performed each of these four steps and created the coding manual, and a secondary researcher helped to refine the manual and assisted with coding ambiguous responses.

The coding manual was used to thematically organize quantitative responses

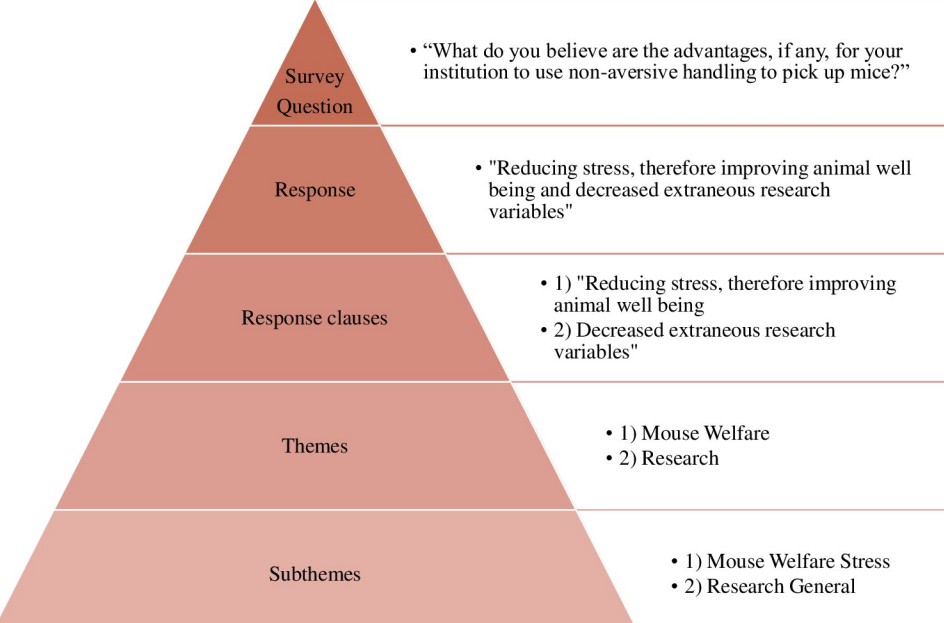

**Fig 2. Coding manual analysis terms and organization.** Graphic outlining the composition of the coding manual for each survey question. Each qualitative question category has its own coding manual that consists of multiple general themes, and each theme has multiple, more specific subthemes. Each theme and subtheme had a definition, distinct key words and example quotes to assist with its understanding. The right side of the graphic contains an example using one of the qualitative survey questions.

Advantages) and created a rough draft of potential themes. An initial coding manual was developed that included the theme abbreviation, proper name, definition, key phrases, and representative quotes related to the specific code (**Fig 2**). Subsequently the data was re-read and formally coded based on the created manual. The coding manual was refined throughout this process via discussion of ambiguous responses with an additional researcher (MRL). Inter-rater reliability was assessed by having a third individual code a random 20% of the qualitative data. They were trained by LEY using the same training process and coding manuals LEY used, and ambiguities were resolved via discussion.

During coding, each participant's response was broken down into its different grammatical clauses; each clause was assigned the appropriate themes independently. There was no limit to the number of themes per response. For example, the response "Not wanting to harm any animals or damage any studies being done with the animal" was coded as the theme <u>Mice</u> and the theme <u>Research General</u>. The theme <u>Mice</u>, was defined as any response detailing a problem with the mice leading to a potential barrier.

Additionally, the coding manual contained main themes and subthemes. For example, the main theme of <u>Mice</u> had three possible subthemes: 'jumpy', 'novel' and 'negative'. Responses were coded as a subtheme when appropriate (e.g., "Not wanting to harm any animals. . .", excerpt from the previous example, would be the theme <u>Mice</u> and subtheme 'negative'). However, if a subtheme was unclear, they were coded only with the header theme. Responses that were non-comprehensive were coded as ambiguous.

Themes were noted as "misconceptions" if they clearly were not supported by published literature and real-world practice. For example, several participants indicated that it would be difficult to use refined handling because they would ultimately need to restrain the mouse by

the tail to conduct procedures. However, by our definition, refined mouse handling only refers to the method used to pick the mouse up out of the cage. Furthermore, multiple research studies have indicated it is beneficial and compatible with common procedure [6, 7, 8, 16, 18]. Additional "misconceptions" are addressed in the discussion.

Ultimately, we calculated the prevalence of each theme by taking the number of participants whose response was coded as a theme for a particular question, divided by the total participants that responded to the question. For example, 65 individuals referenced Mice out of 181 respondents to the 'barriers' question category, yielding a 32% prevalence of Mice as a barrier.

## Results

### Demographics

A total of 261 participants were included in the study with detailed demographic information in **Table 1**. This is very roughly an estimated response rate of 2.6% of eligible participants reached via recruitment (261/10,000). Participants worked in a wide range of roles including veterinarians (26%), caretakers (15%), and managers (15%). Nearly half have either a graduate or veterinary degree (46%). Both academic institutions (61%) and industry (25%) were well-represented. Most participants were located in the United States (78%). Finally, participants

**Table 1. Demographic and work information for study participants (N = 261).**

| Role | N | % of Total |
|---|---|---|
| Veterinarian | 69 | 26% |
| Manager | 40 | 15% |
| Caretaker | 40 | 15% |
| Veterinary Technician | 26 | 10% |
| Researcher | 22 | 8% |
| Research Technician | 18 | 7% |
| Trainer | 16 | 6% |
| IACUC | 14 | 5% |
| Other | 16 | 6% |
| **Highest Education** | | |
| Graduate or Veterinary Degree | 118 | 45% |
| Bachelor's degree | 100 | 38% |
| Associate Degree | 24 | 10% |
| High School Diploma | 16 | 6% |
| No answer | 3 | 1% |
| **Institution Type** | | |
| Academic | 160 | 61% |
| Industry | 66 | 25% |
| Contract Research Organization | 17 | 7% |
| Government | 11 | 4% |
| Other | 7 | 3% |
| **Location** | | |
| USA | 205 | 79% |
| Europe | 24 | 9% |
| Canada | 23 | 9% |
| Other | 9 | 3% |

Note that the percentages are rounded to the nearest full percent so the total percentages may not add up to 100.

**Table 2. Number of mouse cages.**

| # of Cages | # of Institutions |
|---|---|
| 0–1,000 | 33 |
| 1001–5000 | 24 |
| 5001–10,000 | 9 |
| 10,001–20,000 | 9 |
| 20,001–60,000 | 7 |

The number of mouse cages per range reported by representatives from 82 institutions.

were a median age of 39 years old (40 +/- 11) and had been working with mice for median of 11 years (14 +/- 10; n = 258).

## Benchmarking prevalence

**Institutional practices.** Analyzing only one response per institution found the following results across 123 institutions. Participants from 82 institutions reported the number of mouse cages which ranged widely from 2 to 50,000 with a median of 2000 (**Table 2**). In total, survey responses represented 554,322 cages of mice.

Participants from 103 institutions reported the approved methods for picking up mice from their cages (**Fig 3**). Only 5% of institutions allowed <u>only</u> refined methods for picking up mice (e.g., tunnel handling and/or cupping should be used, tail handling is prohibited). Conversely, 18% of institutions allowed only tail handling and/or forceps handling (e.g., tunnel handling and tail handling were not allowed). In terms of allowed techniques, 92% of institutions allowed tail handling, 75% allowed cupping, 68% allowed tunnel handling, 48% allowed tunnel handling, 47% allowed forceps, and 5% allowed scruffing. Of the institutions that

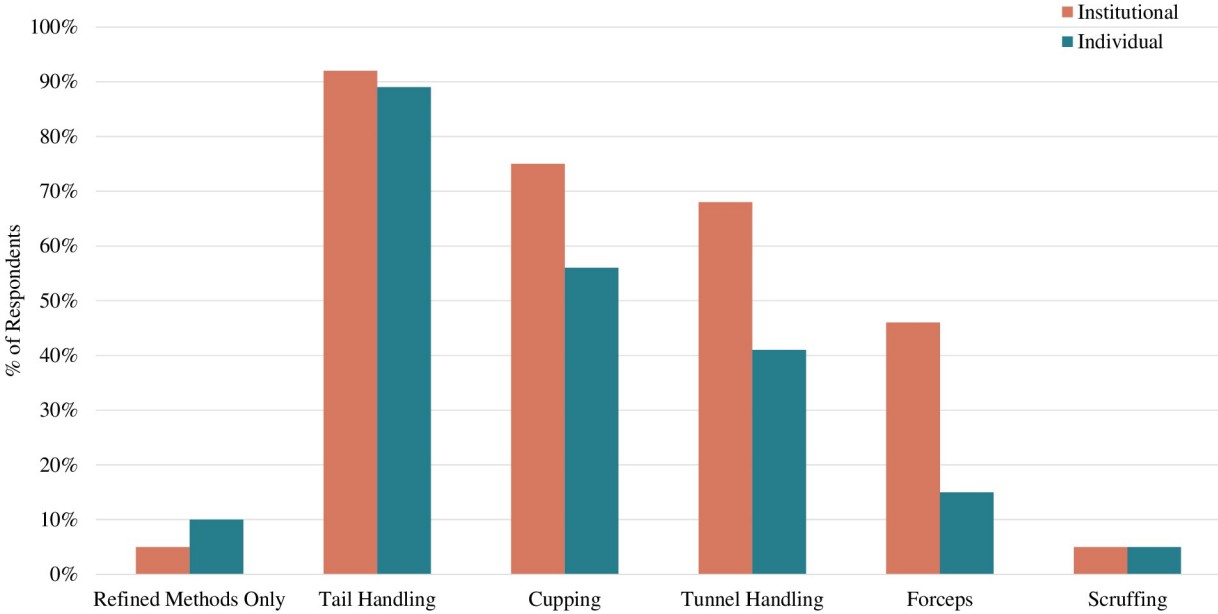

**Fig 3. Handling methods for individuals and institutions: Few institutions or individuals use only refined methods to handle mice.** Laboratory animal personnel were asked to report the approved methods used to handle mice on behalf of their institution (n = 103) and the methods used by themselves personally (n = 215).

allowed forceps handling (n = 40), most were allowed to use forceps on the tail (70%) versus the scruff (29%).

Considering only participants who reported the number of mouse cages and approved handling methods, this indicates that of our sample only 3% of mice in this survey (14,300) are housed at institutions who only permit refined handling, whereas 97% of mice (540,022) are housed at institutions that permit other handling methods.

Participants estimated a median of 10% of mice at their institution are currently picked up with refined handling as the default method (the automatic method used by an individual/ institution if they are not asked to perform a specific type of handling) and predicted a median of 30% would be picked up with refined handling in the next year. Current institutional practices of refined handling were significantly positively associated with control beliefs and norms (Table 3). Institutions from the USA were more likely to report higher implementation. Future institutional practices were significantly positively associated with attitudes, control beliefs, and norms. Again, institutions from the USA were more likely to report higher future intended practice.

On a scale from 1 to 7 (where 1 to 3 indicates negative, 4 indicates neutral, and 5 to 7 indicates positive), on average participants indicated they believed their institution had currently relatively positive attitudes, negative norms, neutral control beliefs, and neutral intent to use refined handling in the next year (Fig 4). Intention to implement refined handling was significantly positively associated with institutional attitudes, control beliefs, and norms (Table 3). Both institutions from the USA versus other, and other institutions versus academic, reported higher intention to implement refined handling in the next year.

**Table 3. Association between institutional refined handling practices, intentions, and explanatory factors.**

| Dependent variable | Independent variables | Estimate | 95% Confidence interval | | Wald χ² | P-value | Generalised R² |
|---|---|---|---|---|---|---|---|
| **Institutional** | | | | | | | |
| **Current practice:** What percentage of mice are currently handled with NAH as default | Intercept | -0.27 | -0.49 | -0.05 | 5.60 | 0.018 | 0.438 |
| | Attitude | 0.02 | -0.01 | 0.04 | 1.00 | 0.317 | |
| | Control Beliefs | 0.04 | 0.01 | 0.07 | 6.03 | 0.014 | |
| | Norms | 0.08 | 0.05 | 0.11 | 26.28 | <0.0001 | |
| | Location (USA vs other) | 0.11 | 0.01 | 0.21 | 4.27 | 0.039 | |
| | Role (manager vs veterinarian) | 0.11 | 0.02 | 0.20 | 5.74 | 0.017 | |
| **Institutional Intention** Intent for future NAH adoption | Intercept | -0.33 | -0.98 | 0.33 | 0.96 | 0.327 | 0.720 |
| | Attitude | 0.53 | 0.43 | 0.62 | 122.10 | <0.0001 | |
| | Control Beliefs | 0.25 | 0.12 | 0.37 | 15.43 | <0.0001 | |
| | Norms | 0.34 | 0.25 | 0.44 | 48.19 | <0.0001 | |
| | Location (USA vs other) | 0.40 | 0.07 | 0.73 | 5.62 | 0.018 | |
| | Institution Type (Academic vs others) | -0.45 | -0.71 | -0.18 | 10.79 | 0.001 | |
| **Future intended practice** Overall, in the next year at your institution, what percentage of mice do you expect to be picked up using default non-aversive handling? | Intercept | -0.43 | -0.62 | -0.24 | 19.14 | <0.0001 | 0.584 |
| | Attitude | 0.04 | 0.01 | 0.06 | 6.93 | 0.009 | |
| | Control Beliefs | 0.06 | 0.04 | 0.09 | 21.23 | <0.0001 | |
| | Norms | 0.10 | 0.07 | 0.13 | 47.63 | <0.0001 | |
| | Location (USA vs other) | 0.10 | 0.01 | 0.19 | 5.02 | 0.025 | |

Regression coefficients and 95% confidence intervals for non-aversive (refined) handling practice determined by Theory of Planned Behavior metrics (Attitude, Control Beliefs, Norms) and work-related attributes.

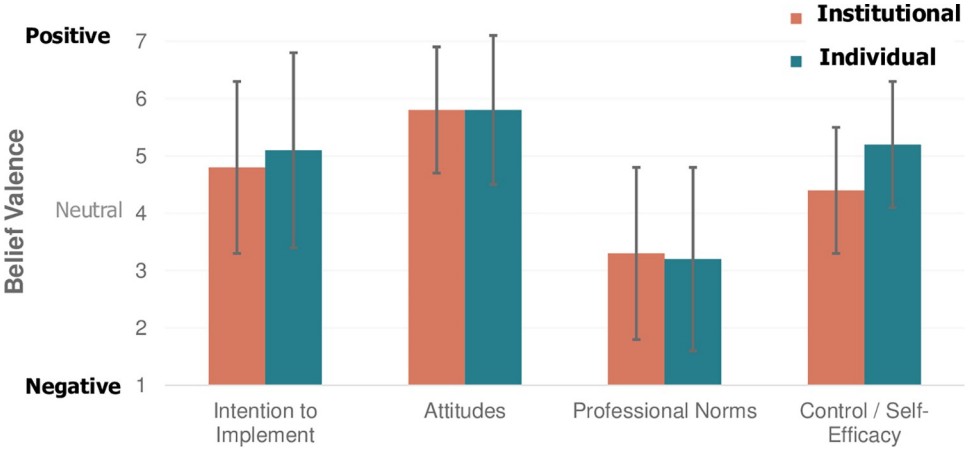

**Fig 4. Beliefs about refined handling.** Laboratory animal personnel were asked to report their beliefs about refined handling on behalf of their institution (n = 93) and for themselves personally (n = 189). All scales were developed from the Theory of Planned Behavior which included intention to implement refined handling in the next year and beliefs about the consequences of (attitudes), professional pressures of (subjective norms), and control over (control / self-efficacy) implementing refined handling. The mean +/- standard error is reported.

**Individual practices.** A total of 215 participants reported the methods they personally use to pick up mice from their cages (Fig 3). Only 10% of individuals used only refined methods to pick up mice. Conversely, 37% of individuals used only tail handling, forceps, and/or scruffing (e.g., they never used tunnel or cup handling). In terms of the techniques used at least some of the time, 89% used tail handling, 55% used cupping, 41% used tunnel handling, 16% used forceps, and 5% scruffed. For those individuals that used forceps (n = 31), the majority used them to pick mice up by the tail (63%) versus the scruff (36%). Participants reported working with a median of 22 cages per day (n = 192 participants) though this ranged widely from 0 to 2000.

Participants estimated currently picking up a median of 18% of cages using refined methods and predicted that would increase to 50% in the next year. Current individual use of refined handling was positively associated with familiarity with refined handling, control beliefs, norms, and knowledge (Table 4). Additionally, USA participants reported higher current practice than individuals from other locations. Future use of refined handling was positively associated with attitudes, norms, and familiarity (Table 4). Additionally, USA participants reported higher intended practice than individuals from other locations.

Overall, most participants were moderately (32%) or very (38%) familiar with refined mouse handling with few being somewhat (18%), slightly (8%), or not at all familiar (2%). The 194 participants that answered the knowledge quiz had adequate knowledge of refined mouse handling with an average quiz score of 75%. The specific percentage of participants answering each question correctly is shown in Fig 5. Generally, participants were knowledgeable about refined handling's benefits (and that these benefits are not counteracted by anesthesia) and that having tunnels in cages is helpful (>80% of participants answered correctly). Participants were less knowledgeable that refined handling is beneficial for all strains, its benefits remain post-injection and restraint, and that tail handling compromises animal welfare (<75% of participants answered correctly).

Participants used a variety of resources to learn about refined mouse handling. This included colleagues (48% of participants), NA3RsC Resources (43%), other websites or conferences (42%), peer-reviewed manuscripts (41%), NC3Rs resources (38%), and technical articles (37%).

**Table 4. Association between individual refined handling practices, intentions, and explanatory factors.**

| Dependent variable | Independent variables | Estimate | 95% Confidence interval | | Wald $\chi^2$ | P-value | Generalised $R^2$ |
|---|---|---|---|---|---|---|---|
| **Individual** | | | | | | | |
| **Current practice** What percentage of mice do you currently handle by non-aversive handling as the default method? | Intercept | -0.32 | -0.52 | -0.12 | 9.63 | 0.002 | 0.438 |
| | Control Beliefs | 0.04 | 0.00 | 0.09 | 4.12 | 0.042 | |
| | Norms | 0.06 | 0.03 | 0.10 | 15.09 | 0.0001 | |
| | Location (USA vs others) | 0.13 | 0.01 | 0.25 | 4.46 | 0.035 | |
| | Familiarity with NAH (Most familiar) | 0.21 | 0.11 | 0.31 | 17.51 | <0.0001 | |
| | Knowledge | 0.01 | 0.00 | 0.03 | 4.04 | 0.045 | |
| **Personal Intentions** In the next year do you intend to use NAH? | Intercept | -0.82 | -1.77 | 0.14 | 2.82 | 0.093 | 0.621 |
| | Attitude | 0.75 | 0.61 | 0.89 | 110.31 | <0.0001 | |
| | Control Beliefs | 0.11 | -0.08 | 0.29 | 1.25 | 0.264 | |
| | Norms | 0.20 | 0.09 | 0.30 | 13.39 | 0.000 | |
| | Role (manager vs veterinarian) | -0.79 | -1.33 | -0.25 | 8.21 | 0.004 | |
| **Future intended practice:** What percentage of mice do you intend to handle using default non-aversive handling? | Intercept | -0.48 | -0.69 | -0.26 | 19.21 | <0.0001 | 0.503 |
| | Attitude | 0.09 | 0.06 | 0.12 | 26.58 | <0.0001 | |
| | Control Beliefs | 0.01 | -0.03 | 0.05 | 0.35 | 0.554 | |
| | Norms | 0.08 | 0.05 | 0.11 | 29.94 | <0.0001 | |
| | Location (USA vs other) | 0.13 | 0.03 | 0.24 | 6.20 | 0.013 | |
| | Familiarity with NAH (Most familiar) | 0.11 | 0.01 | 0.20 | 5.08 | 0.024 | |

Regression coefficients and 95% confidence intervals for non-aversive (refined) handling practice determined by Theory of Planned Behavior metrics (Attitude, Control Beliefs, Norms) and work-related attributes.

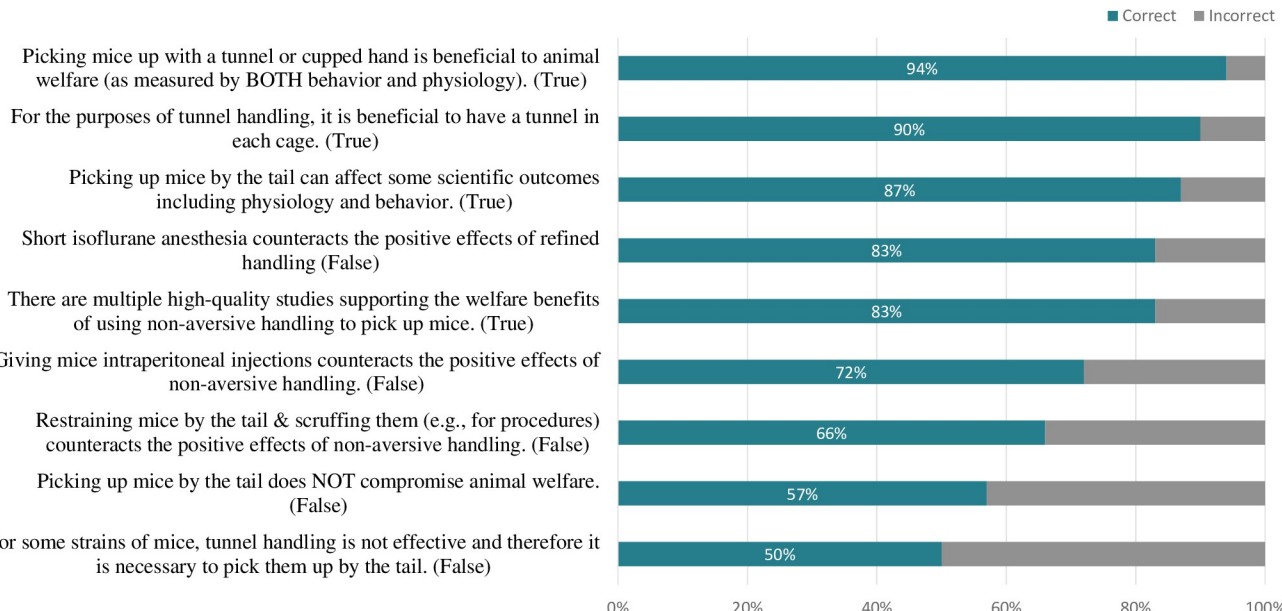

**Fig 5. Participant knowledge of refined handling.** This graph displays the percentage of participants who answered the knowledge quiz (n = 194) and chose the correct answer (based on peer-reviewed literature) to a series of true or false questions. The exact question wording is listed in the figure with the correct answer shown in paratheses.

On average individual participants themselves had relatively positive attitudes, negative norms, and positive control beliefs of refined handling (**Fig 4**). Additionally, most participants (n = 185) agreed that using refined handling was beneficial for animal welfare (88% of participants), science (77%), and people (72%). However, they had only a slightly positive intention to personally use refined handling in the next year (**Fig 4**). Personal intention was positively associated with more positive attitudes and norms (**Table 4**). Additionally, managers had a lower intention to use refined handling than veterinarians. Personal intention was not associated with control beliefs or knowledge.

## Qualitative analysis

Most participants responded to open-ended questions to elicit barriers (n = 181 individual, 71% of participants; n = 173 institutional, 68%) and advantages (n = 179 individual, 70%; n = 175 institutional, 69%). Fewer participants were eligible for and responded to the questions about exceptions to refined handling (n = 131 individual, 51% of participants; n = 61 institutional, 24%) and determining what key factors enabled use of refined handling at their institution (n = 81, 32%). In each section below, we review the key themes in participant responses to each question. Additionally, we will discuss the composite subthemes, for each theme, with their names noted in paratheses. Exact percentages for individual and institutional respondents are provided in parentheses, and a full summary of the results of the qualitative, open-ended questions are included in **S2 Table**.

**Barriers.** Perceived barriers to implementation, for both individuals and institutions, of refined handling were combined into six themes (Mice, Research Procedures, Time, Materials, Personnel and Safety; **Fig 6**). Of note, 65% of participants described individual "barriers" considered to be misconceptions (defined above) such as *Mice*, *Research*, and *Safety*. Additionally,

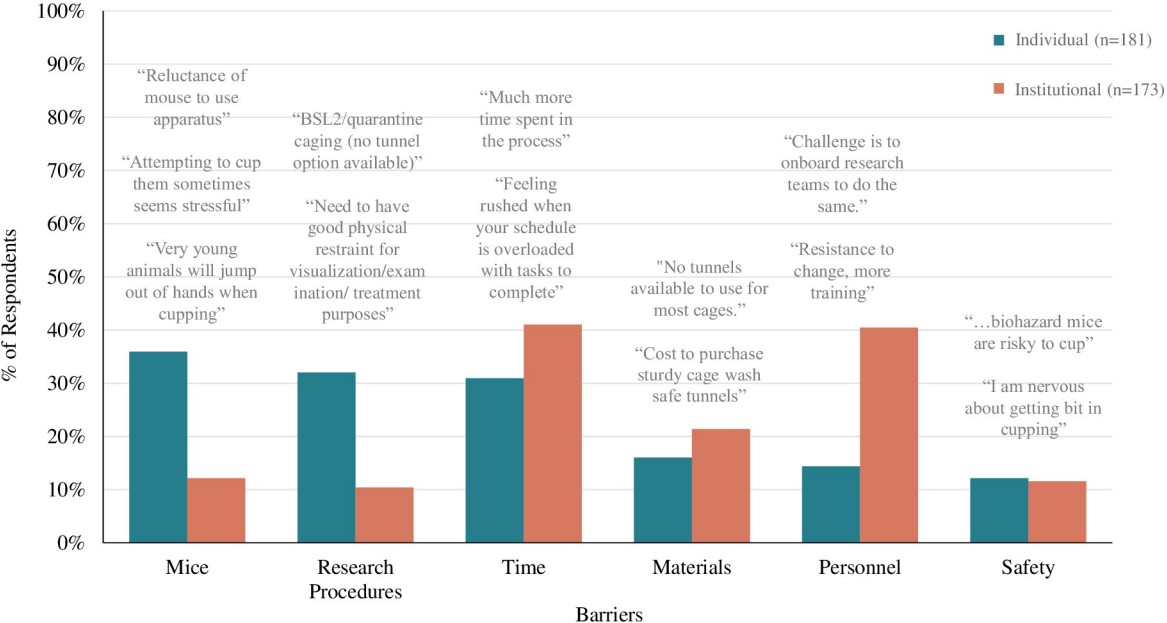

**Fig 6. Barriers to using refined handling were most commonly time, personnel, mice, and research.** Bar graph displays the percentage of respondents whose response contained at least one of the six most prevalent themes to the "Barriers" qualitative survey question ("What factors or circumstances, if any, make it difficult or impossible for you/your institution to use non-aversive handling to pick up mice?"). Left bars indicate responses to individual level survey question (n = 181), right bars indicate responses to institutional level survey question (n = 173).

32% of the situations that participants described as institutional "barriers" are considered to be misconceptions such as *Mice*, *Research Procedures*, and *Safety*.

*Theme 1*: *The mice make it difficult (mice)*. Most commonly, participants described perceived issues with their mice that made refined handling difficult or impossible. These were described by 36% of individual respondents and 12% of institutional respondents, and this theme had several subthemes. Most often participants indicated beliefs that mice were too jumpy (*subtheme name = jumpy;* 20% of individual respondents & 5% of institutional respondents) to use refined handling. One respondent says, "If the mice are jumpy, it may be impossible to use non-aversive handling". Less commonly, participants indicated that mice were not accustomed to this handling (*novel*, 7% & 4%), saying, "when the mice are scared, when first handled on arrival, they might not be used to being handled". Some also believe refined handling would negatively impact the mice (*negative*, 6% & 2%) saying, "Animal care techs are worried about injuring mice if they use a cup (this has occurred at our institution)".

*Theme 2*: *Incompatible with restraint for exams/procedures and research (restraint & research)*. Incompatibility with refined handling, participants find it difficult to perform certain examinations and procedures typically required in a research setting. This was described by 32% of individual participants and 10% of institutional participants. Most often participants indicated concern that refined handling would make restraint for procedures, health checks or injections difficult (*restraint*, 27% & 8%) saying, "usually looking at mice for medical examination and/or treatment and need to restrain animals so they will not bite me during examination and treatment". Less commonly, some participants indicated they were concerned that refined handling would impact laboratory biosecurity (*research biosecurity*, 3% & 2%) saying, "using forceps to pick up infectious or immunocompromised mice is the approved method". A couple participants mention a specific fear of impacting research data collection or damaging studies (*research general*, 2% & 1%).

*Theme 3*: *Not enough time (time)*. Many participants felt that refined handling takes too much time, especially due to increased training requirements, and a lot of mice but too few staff members. This theme was mentioned by 31% of individual respondents and 41% of institutional respondents. Most commonly participants specified their perception that refined handling itself would take too long (*time*, 22% & 24%) stating, "the time we are given to complete tasks and move on to the next study makes techniques that take time difficult to use" and "non-aversive methods can be more time consuming". Less commonly participants specifically mentioned having a large quantity of mice (*quantity of mice*, 8% & 11%) stating, "[refined handling] would be difficult to implement in an institution with 2100 mouse cages." Participants also note a limited number of staff (*staffing*, 3% & 7%) interfered with implementing refined handling stating, "[there are] limited staff to perform husbandry duties, leading to the necessity to work quicker".

*Theme 4*: *Insufficient materials (materials)*. Acquiring the proper materials to conduct refined handling, specifically acquiring tunnels institutionally, was a barrier mentioned by participants. This was described by 16% of individual respondents and 21% of institutional respondents, and participants mention several sub-themes. Participants indicate that they lack access to the materials required for refined handling (*lack materials*, 11% & 13%) stating, "we do not have tunnels yet that can be used in all our animal facility procedures." Less commonly, participants mentioned difficulties keeping the materials properly sanitized (*material sanitation*, 3% & 7%) stating, "[it is] difficult to disinfect tunnels or cups/scoops thoroughly and not have them soaking wet with disinfectant."

*Theme 5*: *Lack of personnel cooperation or training (personnel)*. An additional barrier, mentioned more prominently at the institutional level, was support and buy-in from other personnel. This was described by 14% of individual respondents and 40% of institutional respondents, and the theme contained several subthemes. Most often, participants indicated a

lack of buy-in (*buy-in*, 7% & 27%) from individuals at their workplace to use refined handling, stating "even if our department changed to this method for cage change, health assessments, we would need to convince PI labs to as well." Participants also mention a lack of training (*training*, 6% & 18%) provided to implement it stating, "as for the cupping technique, there is no good reason, apart from the fact that I have not have any opportunity to practice." and "we would need to change our training method." A small percentage of participants also specifically stated a lack of buy-in from officials (*official buy-in*, 4% & 6%) at their place of work (e.g., managers, institution administration) as a barrier specifying, "we have several different affiliate research centres, and it can be difficult to implement new ways of working as they each have to agree to the new SOPs."

*Theme 6*: *Harm to personnel safety (safety)*. A final barrier, described by 14% of participants for both individual and institutional levels, was concern that refined handling would harm personnel safety. More often, participants were concerned that using refined handling would cause injury (*injury*, 8% & 5%), stating that they believed that "cupping can lead to being bit" or that "moving these mice by the tail or with forceps is safest for both the animals and the handler in this instance." Somewhat less often participants specifically called out biosecurity as a concern (*biosecurity, 4% & 6%)*, stating ". . .biohazard mice are risky to cup so I will likely use tail capture until we can get tunnels added to the SOP."

## Advantages

Perceived advantages of implementation, for both individuals and institutions, of refined handling were combined into four themes (Mouse welfare, Handling, Personnel, and Research; **Fig 7**).

*Theme 1*: *Improved mouse welfare (mouse welfare)*. Many participants highlight a primary advantage to using refined handling is its impact on mouse welfare, stated by 78% of individual respondents and 73% of institutional respondents. The majority of individuals specified a

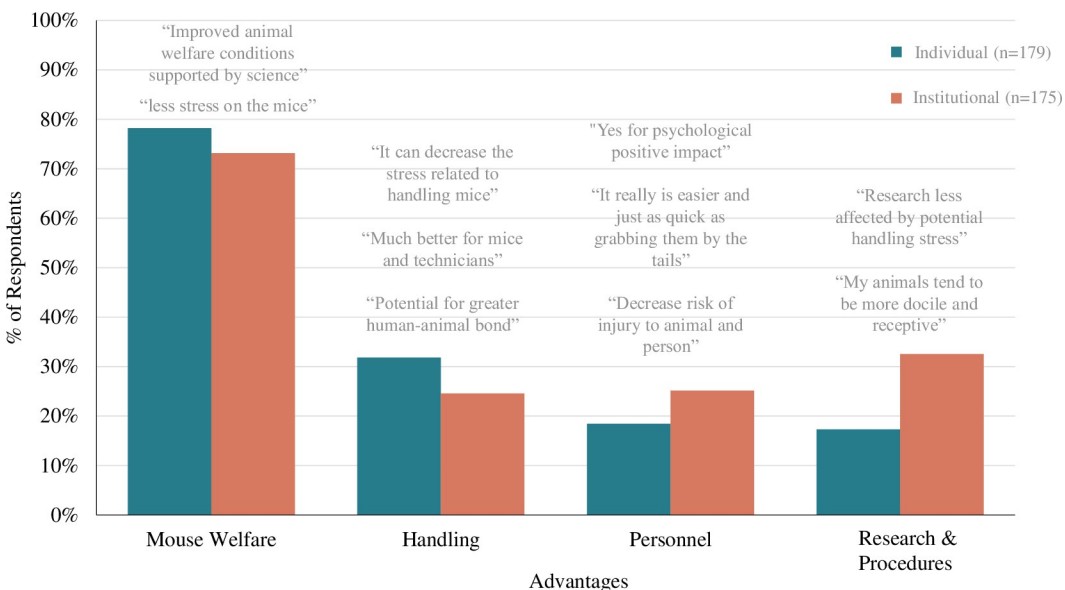

**Fig 7. Advantages to using refined handling.** Bar graph displays the percentage of respondents whose response contained at least one of the four most prevalent themes to the "Advantages" qualitative survey question ("What do you believe are the advantages, if any, for you/your institution to using non-aversive handling to pick up mice). Left bars indicate individual-level responses (n = 179), right bars indicate institutional-level responses (n = 175).

decrease in stress (*subtheme name = stress*, 67% of individual respondents & 50% of institutional respondents), stating "I think this is great welfare for the mice" and "happier mice are more manageable mice, make procedures easier and safer, and are less likely to exhibit stereotypies". A smaller portion of individuals specified either a decrease in mouse injuries (*injury*, 7% & 5%) would be an advantage, stating, "it could make the mice less stressed upon handling and prevent any potential injuries". One participant mentioned this method would be enriching for animals (*enrichment*, 1%) stating, "the rodents are more curious, explorative, and less fearful or stressed upon opening a cage".

*Theme 2*: *Better handling for mice and people (handling)*. Participants also mention the advantages refined handling provides both to the mice and to the people handling them. This was described by 32% of individual respondents and 25% of institutional respondents. Participants specifically indicated handling benefits to the mice (*handling mouse*, 11% & 3%), stating, "yes mice stay calmer, lots of the time they go into their tunnel if that's part of their enrichment and can be passed easily to their new cage without any touching". Participants also indicated it would be better for staff, due to its ease, (*handling handler*, 8% & 7%), or provide simultaneous benefits to both the handler and mice (*handling both*, 4% & 2%) stating, "happier mice and happier techs (I particularly like the interaction when the mice voluntarily climb on to an offered hand".

*Theme 3*: *Benefits the personnel (personnel)*. Participants also mentioned the psychological, emotional, safety and time benefits refined handling provides to personnel and staff. This was described by 18% of individual participants and 25% of institutional respondents. Participants commonly stated the psychological and emotional benefits of refined handling for themselves (*emotion*, 7% & 11%) stating, "improve[d] job satisfaction for the employees" is an advantage. They also indicated the benefits of this handling method were reduced injury risk (*injury handling*, 6% & 6%) stating, "less aversive response in general to being handled by humans = less bite wounds", and improved use of their time (*time*, 4% & 3%) stating, "possibly quicker changing once adjusted ergonomic improvements".

*Theme 4*: *Improves research results and procedures better (research & procedures)*. Some participants mention refined handling provides benefits to research outcomes by reducing the confound of stress due to tail handling. This is described by 17% of individual respondents and 33% of institutional respondents. The majority of participants mentioning research, described an improvement in study results was an advantage of refined handling (*research general*, 11% & 29%), stating, "I agree using non-aversive handling is less stressful to the animal and therefore could have a positive effect on research outcomes." and "better research data". Some participants highlight a specific advantage is the ease and improvement of procedures with this method (*procedure*, 5% & 4%) with participants stating, "we perform many techniques on rodents without anesthesia, so a calm animal is much easier to work with". A few participants also mention improved research due to improved biosecurity (*biosecurity*, 2% & 1%).

**Exceptions.** The exception instances where refined handling would not be used at the individual level were separated into four themes (Mice, Examinations, Safety, Personnel; **Fig 8**). Of note, 52% of participants described individual "Exceptions" considered to be misconceptions such as *Mouse Problems* and *Safety*. Additionally, 36% of the situations that participants described as institutional "Exceptions" are actually considered to be misconceptions such as *Mouse Problems*, and *Safety*.

*Theme 1*: *Mice make it difficult (mice)*. Participants state that they would not use refined handling when they are handling certain types of mice (e.g., jumpy, young). This theme is described by 50% of individual respondents and 36% of institutional respondents and contained several subthemes. Participants stated exception instances to using refined handling

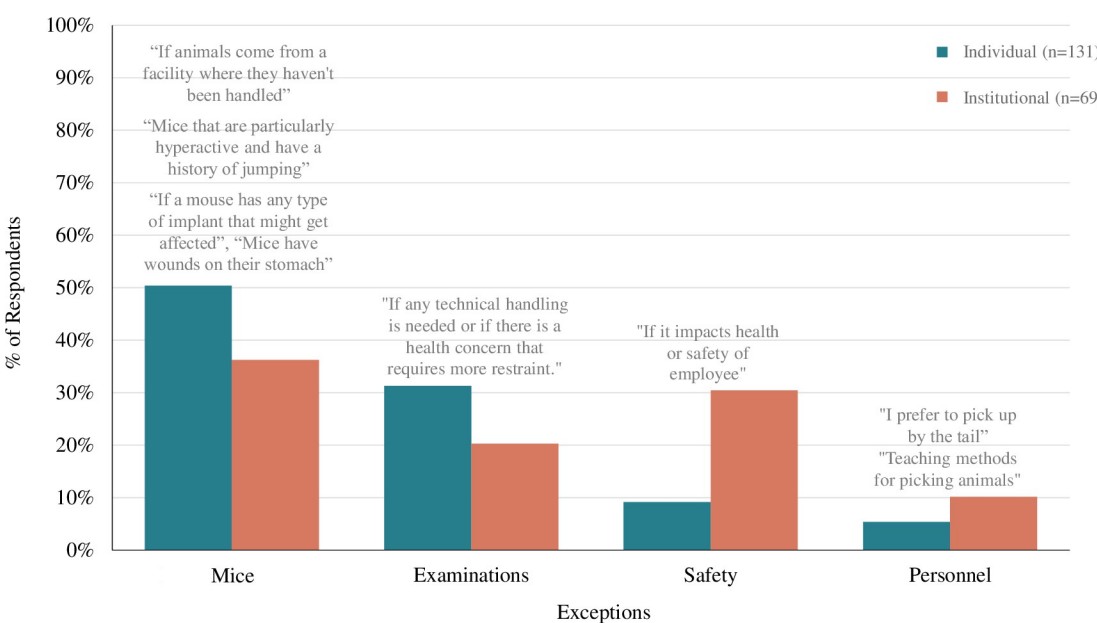

**Fig 8. Exceptions to using refined handling were related to mice, examinations, safety, and personnel.** Bar graph displays the percentage of respondents whose response contained at least one of the four most prevalent themes to the "Exceptions" qualitative survey question ("What, if any, are [the approved] instances/exceptions [for you] to NOT use non-aversive handling to pick up mice?"). Left bars indicate individual-level responses (n = 131), and right bars indicate institutional-level responses (n = 69).

were when they handled jumpy mice (*subtheme name = jumpy*, 22% of individual respondents & 17% of institutional respondents) stating, "'jumpy mice and biohazard mice are risky to cup so I will likely use tail capture until we can get tunnels added to the SOP". Participants also stated exceptions are with aggressive mice (*aggressive*, 12% & 6%), mice unaccustomed to handling (*novel*, 10% & 3%), or mice in vulnerable states such as injured, young, or old animals (*condition*, 9% & 12%). Participants directly state they may not use refined handling " when first handled on arrival [as] they might not be used to being handled", "mice where this causes a more adverse reaction" and when working with "any young adult/adult mouse over 21 days of age" or "neonatal pups, [and] animal too large to fit in tube".

*Theme 2*: *Not compatible with examinations (examinations)*. The second instance where many participants felt they would not use refined handling was when conducting examinations or procedures. This theme was stated by 31% of individual respondents and 20% of individual respondents. Many individuals specified they would not use refined handling when required to perform specific examinations such as research procedures (*research procedure*, 21% & 17%) specifically, "teeth trims, nail trims, subcutaneous injections, IP injections, [and] oral gavage all require scruffing technique". Participants also state "I would need to restrain mice for a complete physical exam" indicating an additional subtheme of veterinary exams or health checks on mice (*veterinary*, 4% & 4%) as exceptions.

*Theme 3*: *Compromises safety and biosecurity (safety)*. Another instance participants state they would not use refined handling was in high biosecurity laboratories, stated by 9% of individual respondents and 30% of institutional respondents. A few participants mention safety in general (*safety general*, 0% & 4%), stating, "if it impacts health or safety of [an] employee or animal" this is an instance they would not use refined handling. Many participants highlight the potential for biosecurity to be compromised (e.g., handling immunocompromised mice, biohazard cages) by using refined handling (*biosecurity*, 9% & 26%) with one participant

stating, "our facility has not adopted tunnel handling so germ-free mice in isolators are typically picked up by the base of the tail using forceps. Outside of the gnotobiotic rooms, I try to use cupping...".

*Theme 4*: *Not the preferred method (personnel)*. Lastly, a few participants mention they would not use refined handling because it is not their personally preferred method to pick up mice. This theme was stated by 5% of individual respondents and 10% of institutional respondents. Participants specified refined handling was not used because other methods catered more towards "personal preference" or it would only be used if "someone just chooses to do it" (*habit*, 2% & 6%).

## Enablers

At the end of the survey, we asked participants whose institutions had switched even partially to refined handling to identify the single most important factor that enabled this switch (**Fig 9**). The most common responses were separated into five themes (Personnel, Mouse General, Materials, Research and Time).

*Theme 1*: *Personnel buy-in, training and leadership (personnel)*. Many participants mention cooperation from personnel (57% of total respondents) enabled them to use refined handling. Participants specified either buy-in/support (*subtheme name = Buy-in*, 31% of total respondents) from other individuals at their workplace, stating "the buy-in from the veterinary care group as we provide training to facility and research staff". They also mention availability of training (*training*, 26%) as an important factor to using it stating, "teaching new starters from the beginning to use methods such as cupping so it become the norm". A smaller number of individuals indicated leadership within their workplace and individuals advocating for refined handling use (*leader*, 9%) was the most important factor enabling them to use it, stating, "the most important factor would be having champions take on this important task"

*Theme 2*: *Better for the mice (mice)*. Many participants note the benefits it provides for the mice, and its effectiveness for specific types of mice enabled them to use refined handling

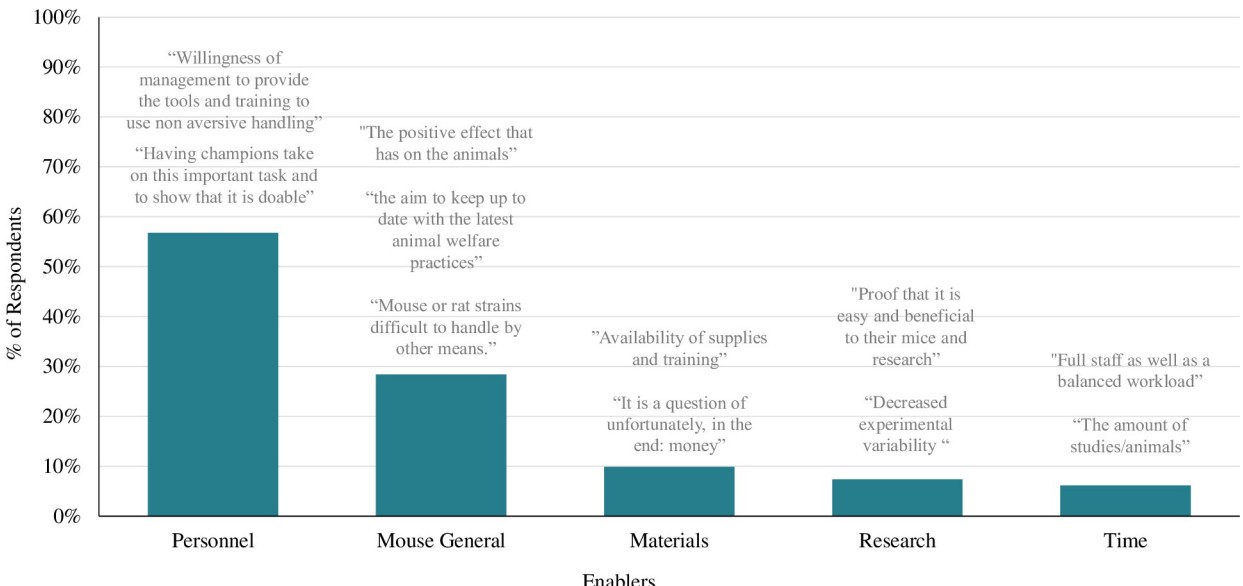

**Fig 9. Enablers to using refined handling were related to personnel, mice, materials, research, and time.** Bar graph displays the percentage of respondents (n = 81) whose response contained at least one of the five most prevalent themes to the "Enablers" qualitative survey question ("What is the single most important factor that has enabled your institution to adopt non-aversive handling methods?").

(28%). Most individuals stated, "a genuine desire to improve animal welfare" (*welfare*, 20%) when handled with refined methods enabled them to use it. Some individuals specified it was necessary to use refined handling to handle difficult (e.g., jumpy, aggressive) mouse strains (*strains*, 4%) stating, "having more species that requires this technique" enabled them to use it.

*Theme 3*: *Availability of materials (materials)*. With tunnel handling being one method of refined handling, this was easier to accomplish for personnel once they acquired the proper materials (10%). Some individuals mentioned that availability of materials needed for refined handling (*lack of materials*, 10%) is important, stating "availability of supplies and training" enabled them to use it.

*Theme 4*: *Benefits for research (research)*. Some participants mention that they are enabled to use refined handling because of the benefits it has for research data (7%). Individuals state they are enabled by, "proof that it is easy and beneficial to their mice and research" and "decreased experimental variability".

*Theme 5*: *Sufficient time to use (time)*. Lastly, a few participants mention having more time (6%), and more staff but less mice, has enabled them to implement refined handling. Specifically, individuals are enabled by a sufficient staff *(staff*, 1%) stating, "it is a question of number of personnel" and the quantity of mice they interact with (*quantity*, 1%) stating, "the [number] of studies/animals".

## Discussion

A total of 261 animal care professionals and researchers from a primarily USA based sample were surveyed to benchmark their current use of and beliefs about refined handling. Most participants claim to be familiar with refined handling and believe it is beneficial for mouse welfare, science and people. However, most people do not frequently use it, do not feel professional pressure to use it, and hold \misconceptions about its use such as believing it is difficult for certain mice and incompatible with restraint/procedures Despite this, participants who reported having more positive attitudes, confidence in implementation, and feeling more professional pressure to use refined handling also reported higher levels of current or future intended practices. Participants from institutions currently using refined handling frequently believed that support from key personnel, training, and leadership were the most important factors that helped them make the switch.

### Current refined handling practices

In May of 2021, participants from this survey reported a low prevalence of refined handling use, despite being familiar with the practice and its benefits. Only 10% of individuals and 5% of institutions use refined handling exclusively. Additionally, our participants estimated only a median of 10% of mice at their institution were handled with refined mouse handling. This is despite the first publication on refined handling in 2010 [8] and as of 2023 at least 22 total publications on the topic (although no formal systematic review exists), [19] and UK reports (60% institutional exclusive use). These findings may indicate that general familiarity with a technique and publications alone do not appear to drive institutional practices.

Despite a lack of default use of refined handling, 55% and 41% of participants reported using tunnel handling and cupping as one of their preferred methods for picking up mice, respectively, which is higher than a previous study [19]. Additionally, tunnel handling and cupping were reported as approved methods for picking up mice at 68% and 75% of institutions, respectively. Finally, participants reported anticipating using refined handling more often in the future.

## Beliefs about refined handling

Laboratory animal personnel overall had very positive attitudes about refined handling with over 70% believing it is beneficial to mouse welfare, science, and personnel. Participants who felt more positively about refined handling were more likely to anticipate higher future implementation of the practice. These findings seem to indicate that refined handling's benefits have been communicated well and should continue to be emphasized to increase implementation. Key studies to cite could include those showing increased voluntary approach [6, 8, 11, 15, 16, 18, 26], decreased anxiety [6, 10, 13, 15, 16, 18, 26, 27], decreased depression-like responses [9, 11, 15] and improved physiological parameters [11, 13, 14, 15, 28, 29].

Despite their positive attitudes to refined handling, participants generally did not feel professional pressure to implement it and 40% indicated personnel were a barrier to implementation due to difficult to get support, buy-in, or training. This barrier has been identified in several prior investigations of barriers to refined rodent handling [19, 20, 30]. However, participants who did feel pressure to implement refined handling reported higher levels of current and future levels of implementation. Additionally, personnel were mentioned as the single most important factor that enabled widespread adoption of refined handling. Therefore, both this survey and prior research [23] further indicates the importance of convincing people and providing training to increase refinement implementation.

Participants were unsure if their institutions could successfully implement refined handling but were somewhat confident that they could implement the practice individually. Interestingly, participants that felt more confident they could implement refined handling were more likely to report higher levels of current practice, but this confidence was not associated with future individual practices. Conversely, participants that felt confident their institution could implement refined handling reported higher future institutional practices, and also reported high current institutional practices. This relative lack of confidence in refined handling could be related to reported barriers that are addressed below.

## Misconceptions, time, & materials required for refined handling

Participants reported several misconceptions about refined handling that may inhibit its wide-scale implementation. First, half of our participants believed that tunnel handling is not compatible either with certain mouse strains or for "jumpy"/aggressive mice. These sentiments were expressed despite the fact that refined handling was developed by researchers studying wild mice, is compatible with multiple strains, and may be most beneficial for helping "jumpy" mice become easier to handle [8]. Second, just under half of participants believed that mouse welfare is not impacted when they are picked up by the tail However, tail handling has been shown in numerous experiments to increase anxiety, stress and depression-like behavior while also decreasing voluntary interaction [7, 8, 9, 11, 15, 24, 25]. Third, around a third of participants incorrectly believed that refined handling is incompatible with common procedures such as restraint for health checks or procedures, matching findings from a prior survey [19] However, multiple studies have indicated that the benefits of refined handling persist after common research procedures (e.g., restraint, injection, anesthesia, oral gavage) [6, 7, 8, 16, 18]. Fourth, although less common, 8% of participants were concerned that refined handling could lead to injury to personnel, particularly bites from mice. This contrasts responses from 6% of participants that stated that refined handling decreases biting incidence. Based on both the knowledge quiz and stated barriers to refined handling, it is evident that further targeted education is needed to combat these misconceptions about refined handling.

A commonly stated barrier to refined handling was the perception that it would take more time to handle mice. A time barrier has been identified in several prior studies of refined

rodent handling [19, 20, 23, 30]. This perception may be due to time required to learn a new technique and early research requiring 30 s plus habituation time for refined methods [8]. However, research now shows that brief refined handling for 2s (compared to tail handling for 2s) is beneficial for both mouse welfare and scientific reliability [7, 17]. Additionally, anecdotally and in 4% of our survey responses participants indicated that once individuals are properly trained, handing takes no additional time or can even be faster. Emphasizing that refined handling takes no more time than tail handling after proper training may help increase implementation.

Another barrier indicated in this study was related to purchasing and sanitizing tunnels. Indeed, anecdotally most facilities that have switched to refined handling as a whole institution have purchased tunnels to be used in all cages—which does require targeted funds. Additionally, these facilities have reported slight increases in time required at cage wash to process these tunnels and properly sanitize them.

### Individual and institutional differences

Overall, there were only a few differences between participant responses about refined handling depending on if they were asked about their individual or institutional practices. Individuals currently used refined methods slightly more and had more confidence about this use than institutions. Barriers to individual practice were primarily related to perceived difficulties with mice, perceived incompatibility with restraint & research, and perception it would take more time. Barriers to institutional practices were primarily related to lack of time, convincing/training people, and access to materials. Addressing all barriers and providing education to both individuals and institutions may be beneficial to increase implementation.

**Targeted education and training are needed to increase refinements.** Taken together, our results seem to indicate that to change implementation of refined handling efforts must be made to change key beliefs about the practice, especially addressing misconceptions and barriers to implementation. Higher implementation levels are seen in people that believe that refined handling is good, feel professional pressure to use it, and feel confident they can use it. Based on this survey, education in venues beyond peer-reviewed publications should seek to emphasize benefits and address common misconceptions about refined handling, especially its application to various strains, compatibility with restraint, and time required.

Regional non-profit organizations may play a key role in promoting refinements by assisting researchers and professionals with disseminating research results and highlighting how their institution overcame barriers. They may be able to help institutions get widespread buy-in with the technique and provide training. In the UK, the National Centre for the 3Rs (NC3Rs) has conducted a variety of targeted educational efforts on refined handling for several years. In turn, much higher levels of implementation of refined handling are seen in the UK. Conversely, the US-based 3Rs Collaborative began targeted efforts to educate local stakeholders about refined handling just after conducting this survey.

### Limitations & generalizability

There were several limitations due to the design and implementation of this survey. With a voluntary, convenience-based survey as used here, it is nearly impossible to have a fully representative sample, and this may be a limitation for the generalisability of our findings. In this case, our sample population was primarily from the USA and therefore our findings may not hold true in other geographical locations. Additionally, our respondent pool may have been biased towards individuals who felt particularly passionate either for or against refined handling. This survey was also cross-sectional, so it is impossible to determine causation for any

associations and/or relationships found. Despite this, past findings show the highly predictive nature of the theory of planned behavior, the methodology we used to investigate relationships between beliefs and intentions. Lastly, the survey was based on self-report, meaning the data collected may be subject to certain biases such as over- or under-reporting of individual's current and planned use of refined handling. Selection bias is specifically a limitation with social media contacts, which was one of the methods used to recruit participants. Despite these limitations, the findings from this survey still provide insight on refined handling prevalence and directions for education to increase implementation. These results may be able to generalize to other areas of the world with relatively similar culture and research animal practices as the United States.

## Conclusion

In conclusion, as of May 2021, refined handling appears to be relatively infrequently implemented in our primarily US-based sample, but increased implementation is anticipated in the future. Although most participants believed refined handling is good, they generally did not feel professional pressure to use it and had minimal confidence in using it. Participants had several misconceptions about refined handling such as perceiving it to be time intensive and incompatible with "jumpy" mice and restraint. Overall, our results suggest that to increase implementation of refined mouse handling, education should focus on improving attitudes, establishing professional norms to use refined handling, and helping improve confidence, potentially by both providing targeted training and addressing common misconceptions. Overall, such interventions have the potential to increase the implementation of refined handling thereby improving the welfare of the millions of laboratory mice used in research worldwide.

## Supporting information

**S1 Table. Refined handling survey data dictionary.** This file contains the exact question and response options provided to participants in the refined handling survey, as well as the corresponding coded values, variable names, and scales.
(XLSX)

**S2 Table. Refined handling qualitative coding manuel.** This file contains one sheet per type of qualitative question asked to survey participants (barriers, advantages, exceptions, and enablers). Each sheet contains the theme name, number and percentage of total respondents (for both individuals and institutions), description of the theme, key phrases, and representative quotes.
(XLSX)

## Acknowledgments

The authors would first like to thank the thousands of laboratory mice used in research worldwide with the hopes that research like this can improve their well-being. We would also like to acknowledge and thank the laboratory animal care teams and personnel who took the time to participate in this survey and provide useful insight, also acknowledging those who continue to champion for refined handling methods. Lastly, we would like to thank the 3Rs Collaborative staff, volunteers, and sponsors for making this research possible, and the participation from the entire refined handling initiative on this project.

## Author Contributions

**Conceptualization:** Donna Goldsteen, Elizabeth A. Nunamaker, Mark J. Prescott, Sally Thompson-Iritani, Sarah E. Thurston, Tara L. Martin, Megan R. LaFollette.

**Data curation:** Lauren Young, Megan R. LaFollette.

**Formal analysis:** Penny Reynolds, Megan R. LaFollette.

**Investigation:** Lauren Young, Donna Goldsteen, Elizabeth A. Nunamaker, Mark J. Prescott, Sally Thompson-Iritani, Tara L. Martin, Megan R. LaFollette.

**Methodology:** Sally Thompson-Iritani, Tara L. Martin, Megan R. LaFollette.

**Project administration:** Sally Thompson-Iritani, Tara L. Martin, Megan R. LaFollette.

**Resources:** Tara L. Martin, Megan R. LaFollette.

**Supervision:** Donna Goldsteen, Elizabeth A. Nunamaker, Mark J. Prescott, Penny Reynolds, Sally Thompson-Iritani, Sarah E. Thurston, Tara L. Martin, Megan R. LaFollette.

**Visualization:** Megan R. LaFollette.

**Writing – original draft:** Lauren Young, Megan R. LaFollette.

**Writing – review & editing:** Lauren Young, Donna Goldsteen, Elizabeth A. Nunamaker, Mark J. Prescott, Penny Reynolds, Sally Thompson-Iritani, Sarah E. Thurston, Tara L. Martin, Megan R. LaFollette.

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
