## [Decision Letter · Decision Letter 0]

24 Apr 2023

PONE-D-23-01633Using refined methods to pick up mice: A survey benchmarking prevalence & beliefs about tunnel and cup handling.PLOS ONE

Dear Dr. LaFollette,

Thank you for submitting your manuscript to PLOS ONE. After careful consideration, we feel that it has merit but does not fully meet PLOS ONE’s publication criteria as it currently stands. Therefore, we invite you to submit a revised version of the manuscript that addresses the points raised during the review process.

This manuscript was well regarded by both reviewers. Please work on addressing the points of confusion and revision recommendations outlined in their positive reviews.  

We look forward to receiving your revised manuscript.

Kind regards,

Cord M. Brundage, D.V.M., Ph.D.

Academic Editor

PLOS ONE

Reviewers' comments:

Reviewer's Responses to Questions

**Comments to the Author**

1. Is the manuscript technically sound, and do the data support the conclusions?

Reviewer #1: Yes

Reviewer #2: Yes

2. Has the statistical analysis been performed appropriately and rigorously? 

Reviewer #1: I Don't Know

Reviewer #2: Yes

3. Have the authors made all data underlying the findings in their manuscript fully available?

Reviewer #1: No

Reviewer #2: Yes

4. Is the manuscript presented in an intelligible fashion and written in standard English?

Reviewer #1: Yes

Reviewer #2: Yes

5. Review Comments to the Author

Reviewer #1: The statement on data availability say it will be available if accepted. Technically this means the data are 'currently' not available. I am not sure if this is acceptable or not. The manuscript contain some minor grammatical errors. See review report to find areas for improvement. I am not familiar with the statistical methods so I am leaving that to other reviewers to decide on the appropriateness of these.

Reviewer #2: This manuscript reports a survey designed to (a) determine the prevalence of refined handling methods to pick up laboratory mice among research personnel working directly with mice, and (b) identify factors that appear to prevent or enable their implementation. Respondents were largely from the USA, with nearly two thirds from academic institutions and one third from industry / contract research organisations. To my knowledge, there are currently no other data available concerning the general use of these methods in North America and individual or institutional attitudes towards their use. This study provides a very valuable contribution towards understanding the current use of an important refinement for the vertebrate species used most frequently in research facilities, and attitudes towards implementing a proven refinement. More generally, it provides a case study to understand knowledge and attitudes towards refinement for animals used in research among the laboratory animal research community.

Overall, the study is well designed, analysed and findings reported quite clearly. There are a few issues that need clarification prior to publication. My review focuses on some general issues before listing some specific issues of phrasing. Points are numbered for convenience of response.

1. There is some confusion in the section of Results benchmarking institutional practices (p18-19) and individual practices (p21) – these are key data to understand, so important to report this as clearly as possible. The survey asks separate questions about what methods are approved for use at an Institutional level, what methods are used by individuals, and what percentage of mice are picked up by non-aversive methods ‘as default’ by Institutions and by individual respondents (it would help to define what ‘as default means here). Fig 3 is labelled current handling methods used by individuals and institutions. There are two different Fig legend titles in the manuscript I am reviewing, but the legend indicates “Laboratory animal personal were asked to report the methods used to handle mice on behalf of their institution”. The text however cites Fig 3 as showing the methods approved at an institutional level for picking up mice from their cages, not what is used (line 350), and Fig 3 as showing the methods personally used by respondents (line 394). While it is ok to show different measures of methods allowed and used in the same figure, this needs to be clearly communicated. It took me a lot of digging to work it out.

2. The study particularly focuses on the percentage of institutions and individuals that use refined methods exclusively and seems to make the assumption that this is the desirable situation and should have high prevalence. However, is there background evidence to support the proposal that refined methods are appropriate in all situations? For example, respondents report that tunnel and cup methods may not be suitable for safety reasons in some situations, or may not be appropriate for use in isolators. I am not familiar with published evidence that refutes this or shows how the methods can be used effectively in isolators and I think exclusive use needs a little more careful consideration, particularly in Discussion (lines 737-747). This is not to argue that the prevalence of current use can be explained by such specific situations, but queries whether exclusive use is the most relevant measure to emphasise rather than the proportion of animals handled.

3. It is also important to report data that allow valid comparisons. In the text, the % of institutions and % of individuals that use refined methods exclusively are given. Then it states “Conversely, nearly all institutions allowed tail handling” (line 353) or “Conversely, nearly all individuals used tail handling” (line 396). This is not a converse (opposite) comparison. How many institutions / individuals used tail handling exclusively by comparison? Then you can compare what proportion used refined methods versus used tail handling, and the estimated proportion of mice handled by tail or refined methods. This is important to understand what proportion of institutions are failing to allow the refined methods as an approved method, or individuals that are completely avoiding their use, rather than using a mixture of methods.

4. Reporting the actual data will help avoid vague statements. For example in line 718-719 of the Discussion “Our results indicated that despite our participants indicating they had high familiarity with refined handling that it is relatively infrequently used.” But what does relatively infrequently used mean? The dictionary definition of infrequent is ‘not happening very often or rare’, but respondents estimated they picked up 18% of mice with refined methods, which doesn’t seem rare, and may be doing this regularly for some mice. A more meaningful statement could be, for example, “Despite 70% of participants reporting moderate or strong familiarity with refined handling, they report that they handled only a median of 18% of mice with these methods.” This is unambiguous. Similarly, see Conclusions line 887.

5. Some of the comments under the qualitative analysis suggest that respondents did not always understand the handling methods to pick up mice that this study specifically focused on, which should be discussed briefly. For example, the comment “usually looking at mice for medical examination and/or treatment and need to restrain animals so they will not bite me during examination and treatment” suggests that the respondent was not aware they could restrain the animal once picked up (line 491-3). The comments that “Animal care techs are worried about injuring mice if they use a cup” and “[it is] difficult to disinfect tunnels or cups/scoops thoroughly” (line 591-20) suggest that some respondents used physical cups for handling rather than cupped hands although this was the focus of the survey.

6. In the generalised linear regression analyses to establish which explanatory variables are associated with implementation of refined handling, it is not clear why specific comparisons are made between particular categories when explanatory variables had multiple categories and how these comparisons were chosen for inclusion (Tables 3 and 4). For example, the influence of location was compared between USA vs all other locations pooled, and the effect of Institution type was compared between Academic vs all others pooled. But the effect of role was compared specifically between Manager vs Veterinarian (not between the most frequent category Veterinarian vs all other categories pooled, while there were as many caretaker respondents as managers). Please provide further explanation and justification in the Data Analysis section to show whether comparisons were chosen to address specific questions of interest, were based simply on the distribution of data (most frequent category vs the rest), or were selected post hoc because there looked to be a difference between specific categories.

7. Explanation of figures should be in the figure legend without an additional title above. There was a problem with formatting of Table 3 in my version, and Tables 1 & 2 are unlikely to meet the formatting requirements of the journal (no boxes). The contrast of colors in Fig 5 was poor and would be difficult for anyone with impairment to see.

8. The paper is quite long and unnecessarily wordy in places. The paper will have maximum impact if people can quickly get to the key points and findings. I would encourage the authors to cut out unnecessary words and phrases, and check punctuation carefully.

Specific comments

9. Abstract line 27-8: ‘yet widespread implementation seems to be low’. Hard to see how widespread implementation can be high or low. Do you mean ‘yet implementation does not seem to be widespread’?

10. Abstract line 28-9: There are other handling methods or combinations of methods that have also been shown to be refinements (for pick up or restraint). Suggest you amend to ‘Refined handling includes ….’

11. Abstract line 31: A hypothesis is normally a proposed explanation for a phenomenon that can be tested scientifically. Instead, these appear to be predictions. What is meant by ‘low’ (need to know that or you cannot assess the prediction). Do you mean compared to traditional tail handling?

12. Abstract line 41: As above ‘low levels’ are not defined. Would be much more meaningful to cite some specific data as in general comment 4 above. Surely it is more useful to understand how many respondents / institutions are using refined methods and on what proportion of animals compared to traditional methods than whether or not they are used exclusively? (particularly if exclusive use is not proven to be a good thing).

13. Intro line 71: Should point out here or elsewhere that other similar methods can also provide a refinement for mice such as picking up mice on a cage ladder (reference 27) or cupping on the hands with massage (reference 13).

14. Intro line 85: You should cite the study that originated the methods here (reference 8).

15. Intro line 110-113: The use of ‘aims’ and ‘objective’ seem to be the wrong way round here. Aim refers to the broad intent, objectives are the steps taken to achieve that.

6. PLOS authors have the option to publish the peer review history of their article (what does this mean?). If published, this will include your full peer review and any attached files.

Reviewer #1: No

Reviewer #2: No

---

## [Author Response · Author response to Decision Letter 0]

14 Jun 2023

Response to reviewers is included at the end of the pdf in a formatted document to assist with ease of reading.

---

## [Editor Report · Decision Letter 1]

19 Jun 2023

Using refined methods to pick up mice: A survey benchmarking prevalence & beliefs about tunnel and cup handling.

PONE-D-23-01633R1

Dear Dr. LaFollette,

We’re pleased to inform you that your manuscript has been judged scientifically suitable for publication and will be formally accepted for publication once it meets all outstanding technical requirements.

Kind regards,

Cord M. Brundage, D.V.M., Ph.D.

Academic Editor

PLOS ONE

---

## [Editor Report · Acceptance letter]

29 Aug 2023

PONE-D-23-01633R1 

Using refined methods to pick up mice: A survey benchmarking prevalence & beliefs about tunnel and cup handling. 

Dear Dr. LaFollette:

I'm pleased to inform you that your manuscript has been deemed suitable for publication in PLOS ONE. Congratulations! Your manuscript is now with our production department. 

Kind regards, 

on behalf of

Dr. Cord M. Brundage 

Academic Editor

PLOS ONE